# Assessing the feasibility of the Virtual Reality Education and Acceptance Protocol among baseball and softball players

Jarad A. Lewellen[1]¤*, Cami A. Barnes[1], Aidan Forget[1], Jeanette M. Garcia[1], D. Jake Follmer[2], Guy Hornsby[1], Hannah L. Silva-Breen[1], Peter R. Giacobbi, Jr.[1]*

**1** School of Sport Sciences, West Virginia University, Morgantown, West Virginia, United States of America, **2** School of Education, West Virginia University, Morgantown, West Virginia, United States of America

¤ Current address: School of Physical Education, Performance and Sport Leadership, Springfield College, Springfield, Massachusetts, United States of America
* jlewellen@springfieldcollege.edu (JAL); peter.giacobbi@mail.wvu.edu (PRG)

## Abstract

Research has supported the use of virtual reality (VR) in sport to train skills such as decision-making and anticipation, as well as aid in injury rehabilitation. Despite this, VR is not commonly used as a training tool in sport. Barriers to its adoption include a lack of understanding, low awareness, risk of cybersickness, and cost. As such, there is a critical need to address these barriers and promote acceptance of VR in sport. The purpose of this single-arm, non-randomized, mixed-methods feasibility trial was to examine the feasibility of the Virtual Reality Education and Acceptance Protocol (VREAP), which was designed by the study's authors to address barriers to VR adoption. While the VREAP is intended to be used in multiple domains, we assessed its feasibility among baseball and softball players. Specifically, we assessed pre- and post-training attitudes toward VR using the Attitudes toward Virtual Reality Technology Scale (AVRTS), which uses the Technology Acceptance Model (TAM) as a guiding framework. Participants ($n = 18$) completed the VREAP, which includes stages of education, acclimation, and application. Exit interviews provided further insights into participant experiences. Results from quantitative and reflexive content analyses demonstrated feasibility of the VREAP based on recruitment and adherence, acceptability, demand, implementation, and practicality. Statistical analyses from the AVRTS revealed significant pre- to post-training increases in overall attitudes toward VR as well as increases in enjoyment, perceived usefulness, and ease of use. Minimal cybersickness was reported. Our findings demonstrate the feasibility of the VREAP among baseball and softball players and show promise for its future research and application.

**Data availability statement:** https://osf.io/8yeqf/?view_only=c7fd7307f-50c4a2db8d6f329727dddd9 All de-identified data and transcript files are available from the OSF repository at the link above. DOI: 10.17605/OSF.IO/8YEQF.

**Funding:** This study was funded by the Stitzel Graduate Student Support Endowment awarded to the first author (JAL) at West Virginia University. The funders had no role in study design, data collection and analysis, decision to publish, or preparation of the manuscript.

**Competing interests:** The authors have declared that no competing interests exist.

## Introduction

Virtual reality (VR) enables users to interact with a virtual environment through a fully immersive (e.g., a head-mounted display; HMD) or a semi-immersive (e.g., projector and screen) system [1]. As the technology progresses, the definition of VR continues to evolve. Le Noury and colleagues [2] used extended reality (XR) as an umbrella term for a spectrum of immersive technology that includes virtual reality (i.e., reality in which the outside world is mostly obscured), augmented reality (i.e., experiencing the real world along with virtual objects overlayed into the field of vision), and mixed reality (i.e., a seamless mix of virtual reality and the real world where users can interact with real and digital objects simultaneously). On the VR end of the spectrum, immersion consists of either animated or 360-degree video using a VR headset. Animated VR uses an environment that is entirely computer generated and promotes a sense of presence in a virtual space while the outside environment is completely obscured. Meanwhile, 360-degree VR includes previously recorded video footage of an environment using a 360-degree camera to immerse users in the virtual space, though they are not able to interact with what they can see as the video is pre-recorded. Though the research on the XR spectrum is expanding, this paper will focus specifically on VR.

As VR technology has developed, researchers from various fields have studied its use in applied settings. For example, the military has used it for exposure therapy for combat-related posttraumatic stress disorder [PTSD; 3] and stress inoculation training [4]. Police departments have used VR as an effective tool in developing more efficient visual search patterns for police officers [5]. The medical field has used VR to simulate surgical procedures [6] and promote self-efficacy in medical students [7]. It has also been used to increase enjoyment, motivation, and engagement while exercising [8].

Within sport, lab-based research has demonstrated the construct validity (i.e., differentiating between expert and novice participants within the virtual environment) and fidelity (i.e., how accurately VR replicates real-world scenarios) of VR. One such study used VR to examine soccer players' ability to stop free kicks and found that experts were better at anticipating the movement of curved kicks and outperformed the novices, suggesting that VR can produce realistic simulations based on level of expertise [9]. Brault et al. [10] similarly examined differences in expert and novice rugby players using VR, where professional rugby players significantly outperformed the novices in each of two experiments examining anticipation of opponent movement.

While these early studies show that VR can discriminate between athletes with different skill levels, other researchers have examined the use of VR for sport-specific skills. A study by Correia et al. [11] used VR to assess decision-making among rugby players and determined that VR effectively recreated training scenarios. Pagé and colleagues [12] conducted a similar study with basketball players where a group that used a VR headset with 360-degree video significantly outperformed both a computer screen and control group on a decision-making task (i.e., using a VR headset to watch plays from a first-person perspective and then finishing the plays with the

correct decision in an on-court post-test). Gray [13] conducted a study in which baseball players across four equal groups were assigned to differing batting practice conditions, two of which used VR, and participants in the adaptive virtual environment group (i.e., difficulty was matched in real-time to the hitter's skill level) improved significantly from pre- to post-training. Meanwhile, Harris et al. [14] examined whether a putting task in an immersive VR environment influenced a real-world putting performance for experts and novices. They found a significant relationship between performance on the VR tasks and the real-world tasks, which indicates that immersive VR can be used to enhance putting training.

In a study examining the impact of VR on motor performance, VR effectively led to more efficient knee movement and more vertical ground reaction force in 20 athletes recovering from ACL injuries [15]. The authors suggested that VR distracted the participants from completing the movements consciously by shifting them to an external attentional focus. VR has also been used to increase motivation in performance, such as when Murray et al. [16] used the presence of another individual within the virtual space to assess motivation and performance among rowers. Participants in the VR groups (i.e., one group rowed individually while the other had a virtual competitor) rowed significantly further than those who did not use VR, and the VR group with the virtual competitor rowed significantly further than the individual VR group. The authors suggested that the use of VR can sufficiently increase motivation and performance, especially when users are presented with a competitive incentive like rowing against a teammate or opponent.

Though the literature is still developing, the early indications are that VR is a valid tool for sport training. In addition to the studies noted above, researchers have demonstrated its usefulness in a variety of sports and among various skill levels. For example, a systematic review by Witte and colleagues [17] included 46 research articles that examined the usefulness of VR in more than 15 sports (e.g., basketball, boxing, football, soccer, table tennis, etc.). These studies supported the use of VR for training skills such as decision-making, anticipation, motor learning, and visual perception. Another systematic review that looked more specifically at studies using randomized controlled trials to evaluate VR in sport found evidence of VR effectiveness in training balance, stability, sprinting, jumping, neurocognitive function, reaction time, and technical skills in studies using football players, basketball players, and physically active adults [18].

Despite the evidence that supports VR in sport, research suggests there are barriers to its adoption in applied sport settings. For example, one study examining VR use in collegiate athletes found that only 6.1% of the 278 participants had ever used VR in any capacity for sport training [19]. When exploring barriers to VR use, Lewellen and colleagues [20] interviewed 14 athletes with prior VR use and found primary barriers to be cost, coach attitudes, lack of awareness, lack of understanding, lack of visibility (i.e., high profile athletes using VR for training), and cybersickness (i.e., when VR users develop symptoms that mimic motion sickness such as headache, disorientation, vertigo, and nausea). These findings were similar to those by Greenhough and colleagues [21], who conducted a study examining soccer player, coach, and support staff perceptions of VR. The survey they administered inquired about acceptance of the technology, VR knowledge, expected performance, social influence, and barriers to use. Further sections inquired about participant VR use or intent to use, depending on whether they had access to it. Among their findings were that players (89%) and practitioners (94%) knew what VR was, though most had never used it in a professional training environment (54% and 70%, respectively). In assessing social influence, participants indicated they would be more likely to use VR if other influential clubs (players = 98%; practitioners = 78%,) or influential others (players = 78%; practitioners = 83%) used it. Practitioners also indicated they would be likely to use VR if players enjoyed using it (98%) or if figures more high-ranking than the player wanted to use it (91%). One of the most noteworthy findings related to barriers to use. The largest barriers were cost of the equipment (92%), limited research within soccer (88%), time available to use it (86%), coach and support staff buy-in (84%), player buy in (80%), space to use the equipment (78%), personnel to operate the equipment (77%), and first impression bias (55%).

A few studies have examined barriers to VR use in sport at the organizational level. One recent qualitative study explored professional European soccer coaches' perceptions of barriers to VR use and opportunities for adaptation [22]. Interviews with six coaches revealed four primary barriers to be perceptions of VR's practicality, lack of empirical evidence

leading to skepticism its value, quality of the technology, and concern that VR could lead to cognitive overload and therefore decreased performance among their players. Coaches suggested opportunities for future VR adaptation were to create team models (e.g., allowing players to experience different situations multiple times or improving team cohesion), use VR for player development (e.g., using the tool for building decision-making skills), use VR for rehabilitation and recovery of injured athletes, and use VR to train isolated incidents (e.g., corner kicks and set pieces). Another study examined practitioner (e.g., coaches and sport scientist) perceptions of VR in elite football and baseball organizations [23]. Participants from both sports viewed improvements in technical and tactical on-field performance to be the most important factor in considering VR use, and that injured and rehabilitating athletes would benefit most from VR. Cost was found to be the biggest barrier to use among both sports, while football practitioners perceived coach and executive approval to be a barrier to adoption.

In domains outside of sport, research has discovered similar barriers. One review of challenges to VR implementation in medical education and treatment noted barriers such as cost, users' attitudes, and VR side effects (i.e., cybersickness) [24]. A recent study examining barriers to VR adoption in therapeutic settings reported four primary barriers: professional barriers (e.g., lack of knowledge, training, time, personal reasons), financial barriers (e.g., costs, cost-benefit-ratio), therapeutic barriers (e.g., clinical applicability, concerns about "real" therapeutic relationship), and technological barriers (e.g., immature technology, cybersickness, no equipment) [25]. These studies show a clear trend across several domains; there is a hesitation to accept and adopt VR despite the evidence that it can be a useful tool.

Though the research on barriers to VR adoption is growing, there is still limited research examining practical strategies for addressing them and therefore increasing the acceptance and application of VR in sport and other domains. One of the most established ways to examine the acceptance of technology is the Technology Acceptance Model [TAM; 26]. The TAM reasons that people will have more intent to use a technology based on their perceived usefulness and perceived ease of use. There is strong evidence supporting the TAM as highly reliable and that perceived usefulness is the strongest predictor of intention to use [27]. In using the TAM to examine the relationship between user experience and VR, Sagnier et al. [28] found that perceived usefulness positively influenced intention to use VR and cybersickness negatively influenced intention to use VR. Additionally, users were less willing to use VR as cybersickness symptoms increased in severity.

Some research has been conducted on VR acceptance in other domains. For example, Wang and colleagues [29] determined that perceptions of usefulness, ease of use, enjoyment, and social influence were all predictors of intent to use VR among graduate medical education trainees. A case study with police officers after VR use demonstrated a link between perceived usefulness, satisfaction, and intent to use [30]. However, few studies have exclusively investigated VR acceptance in sport. One such study conducted by Mascret et al. [31] examined whether the TAM remained valid with VR among 1162 French athletes of various sports at various competitive levels. Participants completed a survey to assess their acceptance of VR before their first use, with measures of acceptance consisting of perceived usefulness, perceived ease of use, enjoyment, and subjective norms. Each of these constructs were found to be predictors of intent to use VR, and there was a small but significant effect for sport level on subjective norms. While these findings support the use of the TAM to assess the acceptance of VR in sport, the study was limited by its survey design, as participants were not exposed to VR, and the researchers only measured responses before initial use. Additionally, the researchers did not examine cybersickness, which is an important consideration given the negative correlation between cybersickness and intent to use VR [32]. With cybersickness being a natural deterrent to VR use, it is important to examine ways to decrease its likelihood so users can experience VR's potential benefits.

One way to help users acclimate to VR is to control the task workload, which was explored by Sepich et al. [33]. They examined the effect of task workload on cybersickness by assigning participants to groups that required varying levels of attentional demand and cognitive workload as they walked through a maze in VR. Low cognitive workloads consisted of gradually progressing through a virtual maze without interacting with its features. High cognitive workload included abrupt turns, rapid movement, and other disorienting features as they completed a memorization task based on animals present in

the virtual environment. They found that the group with the highest cognitive workload during VR was significantly more likely to experience cybersickness than the two groups with medium and low cognitive workload, suggesting that exposing users to a virtual environment with a lower cognitive workload could decrease their likelihood of experiencing cybersickness.

There is a clear need to develop a protocol to limit the risk of cybersickness and increase VR acceptance by providing education and training [20,32]. In response to this need, we developed the Virtual Reality Education and Acceptance Protocol (VREAP), which is outlined below. While the VREAP was designed primarily with sport literature in mind, it is our belief that it could be implemented in a variety of domains that have demonstrated scientific support of VR (e.g., military, medicine, etc.), as the research used to develop the VREAP applies to human behavior regardless of domain. However, a feasibility study for the VREAP is warranted before these assertions can be made. Therefore, the primary purpose of this study was to examine the feasibility of the Virtual Reality Education and Acceptance Protocol (VREAP). Given that the VREAP was designed to be adapted to specific domains and the authors' research domains are primarily sport-related, the current study examines the VREAP's feasibility among baseball and softball players.

## Methods

This study was conducted after approval was obtained from the West Virginia University Institutional Review Board (Protocol #2410050857). All participants provided written informed consent before participating in the study. The reporting of this study follows guidelines for reporting non-randomized feasibility studies [34] and the CONSORT extension checklist and statement for pilot and feasibility trials [35]. A completed checklist is available upon request.

### Research design

Quantitative and qualitative data were collected concurrently as part of this single-arm trial to test the feasibility of the VREAP. This included quantitative and qualitative feasibility data consistent with widely established frameworks [36]. Quantitative data was collected using pre- and post-training surveys, and qualitative data was collected via observation notes from the research team and participant exit interviews at the conclusion of training.

### Participants and recruitment

Because the goal of a feasibility study is not to test a hypothesis, a power analysis to estimate a sample size is not appropriate [37,38]. Rather, proposed pilot study sample sizes should be based on practical considerations, including participant flow, budgetary constraints, and the number of participants needed to reasonably evaluate feasibility goals. Sample size suggestions with these considerations range from 12 to 30 [39]. As such, we sought to recruit 15–20 participants who were (a) at least 18 years old, (b) had competed in organized baseball or softball at the high school level or higher, (c) had no history of severe concussions or traumatic brain injuries, and (d) had limited exposure to virtual reality (i.e., less than 5 exposures over the previous year). Baseball and softball were chosen as the inclusion sports because (a) a baseball- and softball-specific application was chosen for Stage Three of the VREAP, as outlined below, and (b) researchers have determined VR is a useful tool in training pitch recognition [13]. Purposive sampling was used to include participants who met the specified inclusion criteria.

The final sample included 18 participants aged 18–22 ($M = 20.00$; $SD = 1.08$) and consisted of 11 males and 7 females, all of whom self-identified as white. All male participants had experience playing baseball, and all female participants had experience playing softball. Participants' highest level of competition included high school ($n = 11$), college (NCAA; $n = 3$), college (Club; $n = 2$), and recreational ($n = 2$).

### Virtual Reality Education and Acceptance Protocol (VREAP)

The authors of the current study developed the Virtual Reality Education and Acceptance Protocol (VREAP) based on previous research about overcoming barriers to VR use, limiting cybersickness, and increasing acceptance of VR, as

outlined below in each stage. The VREAP consists of three stages: education, acclimation, and application. See Fig 1 for a visual representation of the VREAP stages. Though the VREAP was developed to be applied to several domains, it was developed primarily with the literature on VR in sport in mind. As such, the stages outlined below first explain the general context of each stage followed by how we adapted it to the sport context of the current study. The full protocol is available as a supporting information file (S1 File).

In Stage One (i.e., Education), users are given an 8–10 minute presentation with images and video that educate them on the basic functions of VR and its potential uses in their domain (e.g., sport, medicine, etc.) so that they begin the training with a similar baseline of understanding. This stage is based on previous research demonstrating a positive relationship between educational training and technology adoption [40]. In a sport context, previous research has also shown that athletes might be more willing to use VR if they understand how it can be applied to their sport and if elite athletes in their sport endorse or use it [20]. Because the current study adapted the VREAP to assess feasibility among baseball and softball players, the content of the presentation in Stage One included sport-specific examples and photo evidence that elite athletes in the participants' sports are actively using VR.

Stage Two (i.e., Acclimation) consists of two acclimation periods, each lasting approximately 10–15 minutes with 10–12 minutes between periods for recovery. This structure is based on two factors: 1) studies have shown reduced VR cybersickness in 15-minute exposures [33,41] and 2) previous research has shown that 10–12 minutes between acclimation periods should provide ample time for recovery if cybersickness occurs [33,42]. Each acclimation period progresses in cognitive workload and consists of immersion in a game or application within the VR headset. The first application is TriptoVR, an app that allows the user to explore various geographical locations around the world using video recorded with a 360-degree camera. Users complete the Tour of Venice feature, which lasts approximately eight and a half minutes. During this time, they remain seated in a chair, as the video includes minimal movement with no tactile involvement to allow the participants to begin acclimating to the visual component of VR with limited cognitive workload. The second application is First Steps, an application that is included on all Meta Quest VR headsets. The app is designed as a tutorial to acclimate users to the virtual experience and increase the cognitive workload as the user progresses through the application, including kinesthetic and tactile interactions with the environment. This stage was not altered for the purposes of the current study.

In Stage Three (i.e., Application), users are provided an opportunity to apply VR to their specific domain, thus increasing their perceived usefulness, which has been demonstrated as a predictor of intent to use VR [31]. Participants in this stage are given up to 15 minutes of free time to use a domain-specific application or interact with a domain-specific 360-degree video. In this stage, the user or practitioner implementing the protocol should choose an application that has

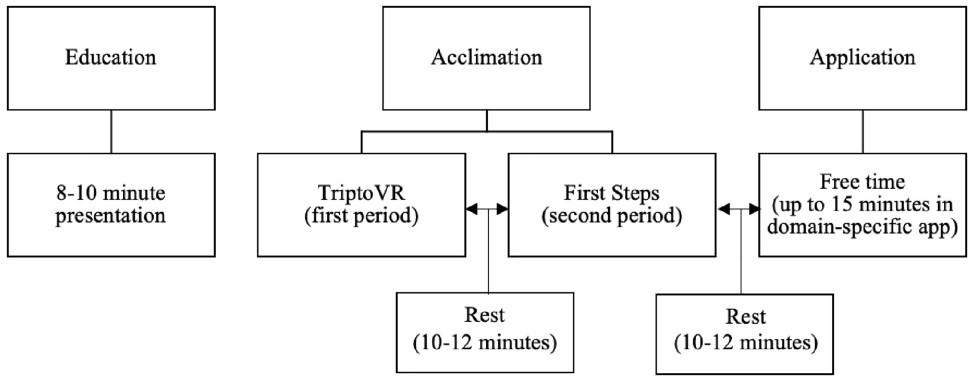

**Fig 1. Virtual Reality Education and Acceptance Protocol.**

demonstrated effectiveness for its domain or use a recorded 360-degree video tailored to the user's needs. For the current study, Stage Three consisted of up to 15 minutes to use the application Win Reality, which is a baseball and softball-specific application designed to train pitch recognition and decision-making. Participants were placed in the batting practice feature where they could use the controllers to simulate swinging a bat. To reflect real-world practice, they were allowed to choose which pitchers and pitches they saw during their batting practice period. They were also encouraged to participate in a minigame that reflected real-life game situations.

## Measures

**Informed consent and screening survey.** Prospective participants were provided with a link to a Qualtrics survey where they could provide written informed consent. The link then led them to a survey that assessed eligibility based on the above criteria and gathered demographic information (i.e., age, gender, ethnicity, competition level, and primary sport).

**Feasibility.** To assess feasibility, we measured recruitment and adherence rates as well as four of the areas of focus for feasibility studies outlined by Bowen and colleagues [36]: acceptability, demand, implementation, and practicality. See Table 1 for specific feasibility questions, outcomes, and measures according to each area of focus. Because this study is believed to be the first of its kind, there are no other studies examining similar feasibility measures with a similar population. As such, we did not establish benchmarks of feasibility for our measures. Rather, we intend that this study will provide benchmarks for future research in this area.

*Recruitment and adherence*: The number of individuals who enrolled in the study (i.e., provided informed consent and met all inclusion criteria), scheduled participation, and completed the study were recorded. Recruitment feasibility was defined as the ability to recruit between 15 and 20 participants. Participant adherence was assessed by attendance, engagement in the training protocol, and the number of measures completed. Attendance was measured by how many of the scheduled participants reported to their time slot. Engagement was assessed by observing how much of the 15-minute free period participants used in Win Reality (i.e., the application stage of the VREAP) and documenting any observable indicators of engagement with the app (e.g., immersion, self-talk, enjoyment, etc.). If participants did not use the entire 15 minutes, they were asked to provide a reason why they discontinued use. Based on Ross-Stewart et al.'s [43] study in which participants had an 81.5% adherence rate to an imagery program using VR, we considered 81.5% adherence to demonstrate feasibility.

**Table 1. Feasibility areas of focus, questions, outcomes, and measures.**

| Area of Focus | Feasibility Questions | Feasibility Outcomes | Feasibility Measures |
|---|---|---|---|
| Acceptability | Is the VREAP judged as suitable, satisfying, or attractive to program recipients? | • Satisfaction<br>• Treatment credibillity<br>• Suggestions for change | • Observation notes<br>• Exit interviews |
| Demand | Is the VREAP likely to be used? | • Expressed interest or intention to use<br>• Perceived demand | • Observation notes<br>• Exit interviews |
| Implementation | Can the VREAP be successfully delivered to participants? | • Success or failure of execution<br>• Amount, type of resources needed to implement<br>• Factors affecting implementation ease or difficulty | • Number of technical difficulties<br>• Number of participants who experience adverse reactions<br>• Observation notes<br>• Exit interviews |
| Practicality | Can the VREAP be carried out with intended participants using existing means, resources, and circumstances and without outside intervention? | • Efficiency, speed, or quality of implementation<br>• Ability of participants to carry out training activities | • Average time of training session<br>• Number of dropouts<br>• Outside materials needed<br>• Observation notes<br>• Exit interviews |

**Impact of the Virtual Reality Education and Acceptance Protocol.** *Virtual Reality Sickness Questionnaire*: Based on the Simulator Sickness Questionnaire [44], the Virtual Reality Sickness Questionnaire [VRSQ; 45] was designed to measure the extent to which VR headset users experience cybersickness during or after use. The subscales measure oculomotor (i.e., general discomfort, fatigue, eye strain, difficulty focusing) and disorientation (i.e., headache, fullness of head, blurred vision, dizzy, vertigo) scores. There is strong reliability for each subscale: oculomotor ($\alpha = .847$) and disorientation ($\alpha = .886$). Items are measured using a 4-point Likert scale ranging from 0 (i.e., "not at all") to 3 (i.e., very"), with higher scores indicating more (i.e., worse) cybersickness.

*Attitudes toward Virtual Reality Technology Scale*: The Attitudes toward Virtual Reality Technology Scale [AVRTS; 46] is a 22-item scale used to assess a person's attitudes toward VR. Overall reliability for the scale is $\alpha = .910$, and the subscales of perceived ease of use ($\alpha = .858$), perceived usefulness ($\alpha = .857$), and enjoyment ($\alpha = .919$) also have strong reliability. All items are scored using a 5-point Likert scale that ranges from "strongly disagree" to "strongly agree." Higher scores indicate more positive attitudes towards VR.

*Exit interviews*: Exit interviews consisted of open-ended questions intended to gather information about feasibility and participants' attitudes toward VR (e.g., insights into their experiences and views, intent to use VR in the future, etc.). Questions were guided by the study's purpose. They included items such as, "What are your thoughts about using this VR protocol for your sport?" and "Did your experience impact your willingness to use VR for your sport? Why or why not?" Interviews lasted an average of 4 minutes and 18 seconds.

*Observation notes*: Members of the research team wrote down observations for each participant. The standardized observation sheets used by the researchers included space to document measures related to acceptability (e.g., how many minutes participants used Win Reality), implementation (e.g., number and description of technical difficulties), and practicality (e.g., start and end time of training session). There was also space to note subjective observations of participants' behaviors during training. The standardized observation notes were developed in part based on previous research that has used similar feasibility measures [e.g., 47,48].

## Procedures

After receiving approval to conduct this study, potential participants were recruited over a 5-week period (02/12/2024–10/01/2025) to complete the screening survey and provide written informed consent. Those who passed the screening were enrolled in the study and emailed to schedule a day and time for participation in a private classroom on the authors' campus. To ensure efficient data collection, and because social norms may positively impact attitudes toward VR [21,31], participants were run in groups of two to four. Participants completed exit interviews after Stage Three, and members of the research team collected observational data using the standardized observation notes. To examine potential changes in cybersickness symptoms and attitudes toward VR, the VRSQ and AVRTS were administered upon arrival (i.e., pre-training) and after Stage Three (i.e., post-training).

Given that exposure to VR carries a minimal risk of adverse reactions (i.e., cybersickness), we took precautions to ensure that such occurrences were handled appropriately. Prior to immersion, participants were instructed to remove the VR headset if they felt unable or unwilling to continue. We also provided a designated recovery area within the testing room if participants needed extended recovery time or extra attention. This area included chairs, bottled water, and saltine crackers to aid in recovery, as well as a trash bin in case of nausea or vomiting. Members of the research team were briefed on emergency protocol in case a participant experienced any adverse reactions during or after exposure to VR. Participants who completed all study measures were provided a $20 Amazon gift card at the conclusion of testing.

## Data analysis

Data gathered to answer feasibility questions in Table 1 were computed as frequencies and percentages. Descriptive statistics were computed for all measures in the current study related to measures of cybersickness and attitudes toward

VR. A Shapiro-Wilk test revealed all VRSQ data were non-normally distributed ($p < 0.05$), and boxplots revealed outliers in the AVRTS data. Therefore, we conducted Wilcoxon signed-ranks tests for the overall scores on the VRSQ and AVRTS to examine potential efficacy of the VREAP among all participants and independent gender groups. The qualitative interviews were analyzed using the six stages of reflexive content analysis [RCA; 49]. Two members of the research team first independently coded all 18 interviews using an inductive approach to allow various codes to emerge. The researchers independently generated 198 and 230 initial codes. They then met with a third member of the research team to discuss and revise codes over the course of two meetings. During these meetings, the three coders discussed each of the initial codes and determined which of them could be combined under the same code. These discussions resulted in 66 well-defined codes. The three coders then deductively assigned each code to a subcategory and category based on the study's feasibility outcomes (e.g., satisfaction, expressed interest or intention to use, etc.) and areas of focus (e.g., acceptability, demand, etc.), as well as the TAM for attitudes toward VR (e.g., perceived usefulness of VR after completing the training). There was minimal debate amongst the three coders on which codes were assigned to each subcategory. However, when disagreements arose, each coder presented their rationale, and the coders considered each option until a consensus was reached. A fourth member of the research team conducted an audit of five random interviews to assess whether the existing analysis was sufficient. The first three coders had been previously trained in qualitative data analysis and had experience coding for previous research studies. The fourth coder had no prior coding experience but was being trained in qualitative methods by the first author, an experienced qualitative researcher. This constant collaboration and discussion among the research team enhanced the quality of the analysis. We also conducted a content analysis of the observation notes to determine if any researcher notes reflected any codes, subcategories, or categories that emerged from the interviews. To do this, the first two authors compiled the qualitative data from the observation notes into a master spreadsheet and generated initial codes from the data. These codes were then assigned to the categories created from the RCA. No new codes were identified through the observation notes. The triangulation of the survey results, interviews, and observation notes enhanced the trustworthiness of the analysis [50].

## Results

### Feasibility

**Recruitment and adherence.** Strategies for recruitment included contacting competitive NCAA and NCAA Club (i.e., student-run teams that compete against club teams from other universities) baseball and softball teams at the researchers' institution, sharing recruitment flyers with four online courses taught by members of the research team, and the lead researcher attending five additional in-person courses to share information about the study and provide the link to enroll. Over a 5-week period (02/12/2024–10/01/2025), 163 individuals opened the screening and informed consent survey. Of these, 26 (16%) provided written informed consent, met all inclusion criteria, and were enrolled in the study. Enrolled participants were then emailed to gauge availability and sign up for a day and time to participate.

Fig 2 provides a CONSORT flowchart for this single-arm, nonrandomized feasibility study. Of the subsequent 19 participants who responded to emails and scheduled their participation, 18 attended their time slot and completed the VREAP, resulting in a 95% attendance rate. We stopped recruitment and collection once we obtained enough participants to reach our goal based on previously established guidelines [39]. Participants were run in five groups over the course of eight days. Regarding engagement, 17 of 18 participants (94%) used all 15 minutes of free time during the application stage. Participant 14 only used 13.5 minutes and explained that he stopped early because he was "struggling to hit" in the app and did not want to use the remaining time to begin a new batting practice session. Other indicators of engagement included multiple participants using motivational self-talk during their use of Win Reality (e.g., "There you go; good hit." Participant 6) and excitement to use different functions of the app. For example, participant 16 expressed excitement in facing a pitcher who threw 100 miles per hour. Additionally, all measures were completed by all 18 (100%) participants.

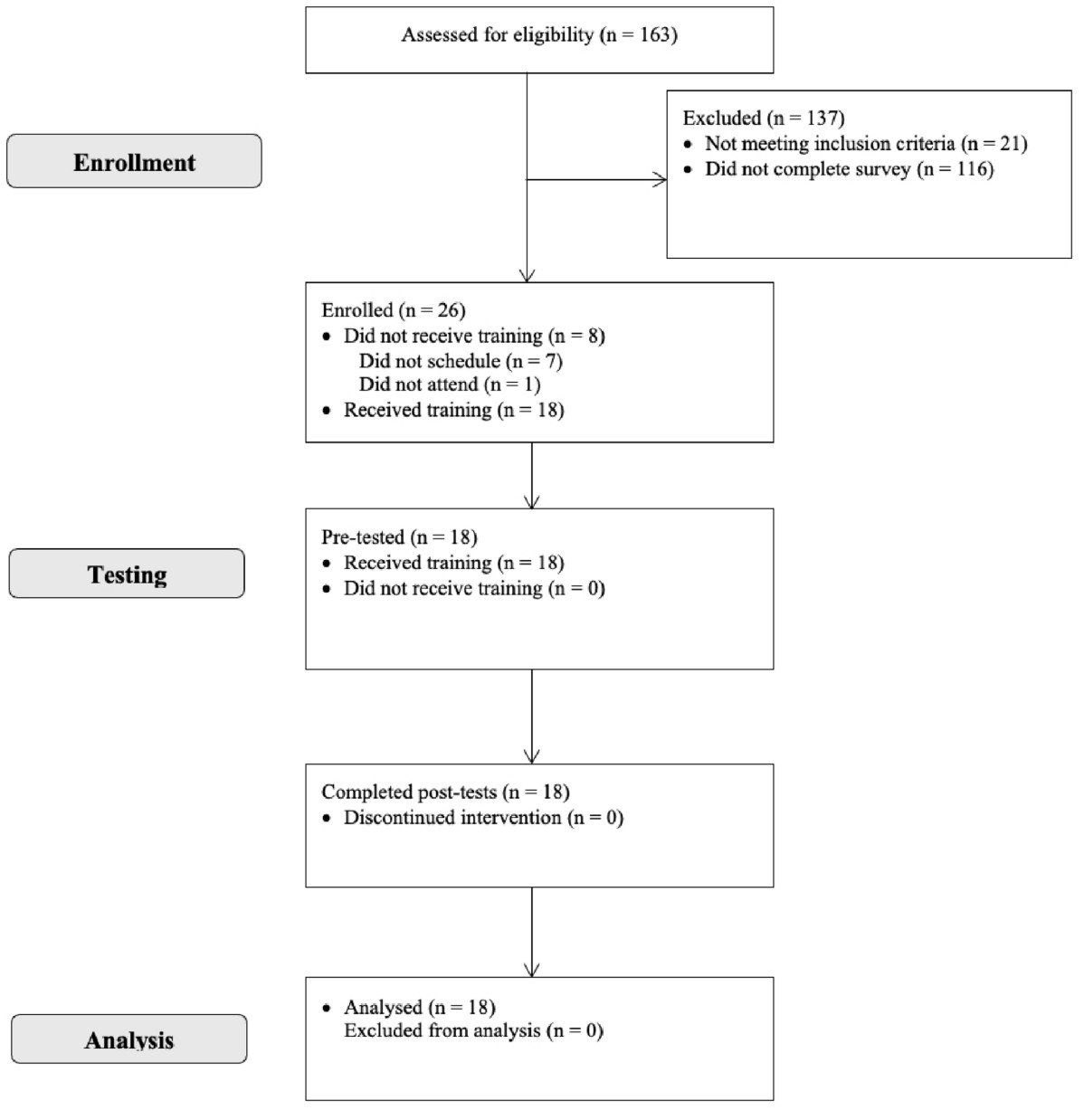

**Fig 2. CONSORT flow diagram for single-arm trials.**

**Acceptability.** Participant interviews revealed high levels of satisfaction, acceptance, and treatment credibility. All 18 (100%) of the participants indicated their satisfaction with the VREAP and training experience by using words such as fun, positive, engaging, or helpful to describe it. For example, participant 6 stated, "It was pretty fun. I enjoyed it. Baseball was really fun. Also, the second app [First Steps] we used, I definitely enjoyed it, with the dancing and everything. That was fun".

Seventeen of 18 (94%) participants also indicated that the treatment (i.e., the VREAP) was credible by discussing how it increased their understanding and acceptance of VR and did not result in symptoms of cybersickness. Participant 12, for

example, noted that she "was kind of skeptical about it [before the training], because I never really did VR, and I thought it was more of a gaming thing. I didn't know it's so accessible." Participant 17 explained:

I've only used [VR] once or twice before, and when I used it before, I got really dizzy and motion sick. But I didn't this time, so I didn't think that I would be able to do [the protocol]. But I didn't [get cybersickness].

Eleven (61%) of the 18 participants supplied suggestions for changes in future iterations of the VREAP. The most common suggestion (n = 6) was to allow future participants to use the bat attachment that is available for Win Reality and allows users to swing real bats while being provided with feedback on their swing within the virtual environment. As participant 4 stated, "I think it would have been a little bit easier if I actually had a bat in my hand. With the controller it was a little weird." Other suggestions included more gaming options within Win Reality (e.g., competing against other users; n = 2), shortening the first application (n = 1), switching the order of the first and second apps to "figure out what VR is and how to use it [first]" (Participant 15; n = 1), and choosing a first app that was more interactive (n = 1).

**Demand.** Seven of the 18 (39%) participants expressed an intent to use or perceived demand for the VREAP in applied settings. For example, participant 2 stated, "if [coaches and athletes] haven't used the VR, I would do [the protocol]. Participant 16 elaborated:

I think it's definitely necessary to go through somewhat of a protocol, whether it's the exact same as we did today, or any other sort of introduction…Having a tool like this to get comfortable with is something that I've never had before.

**Implementation.** Six of the 18 participants (33%) experienced minor technical difficulties during training, which included the bat being in the wrong hand within Win Reality (n = 3), controllers not working properly (n = 2), or an app not loading (n = 2). These technical difficulties were addressed and remedied within one to four minutes by a member of the research team. Zero participants noted adverse reactions during training. The primary factors that affected the ease of implementation were comfort within the virtual environment and a participant's previous experience with VR. Two (11%) participants used the words "safe" or "comfortable" to describe how it felt to be in the virtual space. Three participants (17%) noted previous experience using VR that prepared them for what to expect. Participant 6 noted that he is "not usually someone who has issues with virtual sickness." Several participants also noted that they enjoyed the novelty of the new experience, whether they had prior VR experience or not. Participant 15 noted, "I don't have a lot of experience with VR, so that was kind of a new thing. It was a pretty fun and engaging experience." Meanwhile, the primary factors that affected the difficulty of implementation were technical difficulties and participant characteristics. As noted above, there were seven minor technical difficulties across all groups, with participant 2 stating that the "delay wasn't ideal" in his interview. Finally, three participants (17%) wore glasses and had to remove them for the headset to fit on their head, though none of them commented on this in their exit interviews.

**Practicality.** The training sessions ranged from 72 to 81 minutes (M = 77.6 minutes). Factors that contributed to the speed, efficiency, and quality of implementation were a lack of dropouts (n = 0) and no use of outside materials. Participants completed the training activity themselves and only required assistance from the research team when experiencing a technical difficulty. Fourteen (78%) participants commented on the quality of the VREAP. Participant 10 noted that the training "was very well thought out and well planned. [The researchers] had everything set up where I would set it up if I were doing it."

### Cybersickness and attitudes toward VR

In addition to no reported adverse reactions during use, scores on the VRSQ indicated minimal cybersickness. Overall scores (out of 100) increased by 7.6% after training from 1.85 (SD = 2.94) to 1.99 (SD = 2.82), with a negligible effect size

($d$ = .05). A Wilcoxon signed-rank test showed that participants' overall scores on the VRSQ did not change significantly after testing ($Z$ = −0.086, $p$ = 0.931). Oculomotor scores decreased by 12.4% after training from 3.70 ($SD$ = 5.87) to 3.24 ($SD$ = 5.06) with a negligible effect size ($d$ = .08). Only one participant audibly noted discomfort during training, as participant 17 said her head was "feeling heavy" while wearing the headset. Disorientation scores also increased after the training from 0.00 ($SD$ = 0.00) to 0.74 ($SD$ = 2.16). Among males, overall VRSQ scores decreased by 17% after training from 2.65 ($SD$ = 3.37) to 2.20 ($SD$ = 2.82), with a negligible effect size ($d$ = .14). A Wilcoxon signed-rank test showed that males' overall scores on the VRSQ did not change significantly after testing ($Z$ = −0.736, $p$ = 0.461). Among females, overall VRSQ scores increased by 178.3% after training from 0.60 ($SD$ = 1.57) to 1.67 ($SD$ = 3.00), with a small effect size ($d$ = .45). A Wilcoxon signed-rank test showed that females' overall scores on the VRSQ did not change significantly after testing ($Z$ = −0.816, $p$ = 0.414).

Overall scores on the AVRTS (out of 5.00) increased by 21.5% from 3.63 ($SD$ = 0.55) to 4.41 ($SD$ = 0.33) with a large effect size ($d$ = 1.72), and all 18 (100%) participants' scores increased. A Wilcoxon signed-rank test showed that participants' overall scores on the AVRTS changed significantly after testing ($Z$ = −3.725, $p$ < 0.001). Scores for enjoyment increased by 16.2% from 4.08 ($SD$ = 0.67) to 4.74 ($SD$ = 0.33) with a large effect size ($d$ = 1.25). On this subscale, two (11%) participants' scores stayed the same (i.e., 5.00/5.00 pre- and post-training), and the other 16 (89%) participants' scores increased. Perceived usefulness of VR increased by 36.9% from an average of 2.98 ($SD$ = 0.72) to 4.08 ($SD$ = 0.58) with a large effect size ($d$ = 1.68), with all 18 (100%) participants' scores increasing. Finally, perceived ease of use increased by 18.2% from 3.62 ($SD$ = 0.64) to 4.28 ($SD$ = 0.63) with a large effect size ($d$ = 1.04). One (6%) participant's score stayed the same pre- and post-training (i.e., 4.86/5.00), and 15 (83%) of 18 participants' scores increased. Among males, overall AVRTS scores increased by 17.7% after training from 3.79 ($SD$ = 0.45) to 4.46 ($SD$ = 0.29), with a large effect size ($d$ = 1.80). A Wilcoxon signed-rank test showed that males' overall scores on the AVRTS changed significantly after testing ($Z$ = −2.936, $p$ = 0.003). Among females, overall AVRTS scores increased by 28% after training from 3.39 ($SD$ = 0.64) to 4.34 ($SD$ = 0.38), with a large effect size ($d$ = 1.81). A Wilcoxon signed-rank test showed that females' overall scores on the AVTS changed significantly after testing ($Z$ = −2.366, $p$ = 0.018).

The reflexive content analysis of exit interviews yielded three subcategories deduced using the TAM [26]: perceived usefulness, perceived ease of use, and willingness to use VR. All 18 participants (100%) noted their perceived usefulness of VR after engaging in the training, such as when participant 10 noted, "I think it would be very helpful for people specifically with injuries … and still, maybe get into some of those reps and still see where the ball is coming from and pitch recognition." Only four (22%) participants reported perceptions that VR was easy to use after the training. Participant 16 noted "how easy it was to use. I don't see how an athlete could really have a hard time using it. I feel comfortable with using it not even after an hour." All 18 participants (100%) expressed a willingness to use VR for sport training again if given the opportunity. Participant 16 continued, "I would totally [use it again]. If any coach came to me and said, 'we're going to train VR today,' I'd be happy."

## Discussion

This study was designed to examine the feasibility of the VREAP, which was designed to minimize the risk of cybersickness and promote acceptance of VR, among baseball and softball players. Results of the study indicated feasibility related to recruitment and adherence based on benchmarks established in previous studies [43,51]. Acceptability, demand, implementation, and practicality outlined by Bowen and colleagues [36] also supported the feasibility of the VREAP among baseball and softball players. Statistical analyses and interviews with participants also indicated that the VREAP has the potential to limit the risk of cybersickness and positively affect attitudes toward VR (e.g., acceptance). Specifically, the results suggest that completing the VREAP did not significantly increase the likelihood of experiencing cybersickness symptoms, and it significantly increased acceptance of VR.

Sixteen percent of screened individuals were ultimately enrolled in the study, which is lower than previous VR feasibility studies shown in a review of the literature [51]. Several factors could have influenced this number. For example, one inclusion criterion was a history of playing competitive baseball or softball. While this was done intentionally because participants played a sport-specific app in Stage Three, future iterations of the VREAP might include specific apps specific to other sports or other domains that could provide more options for participants from various athletic backgrounds and domains. Even so, a 95% attendance rate over an 8-day period demonstrates that it was feasible to recruit the desired sample for this study. Participants also showed high engagement and completed all study measures, which suggests feasibility for the VREAP.

Participants also demonstrated high acceptability, as they found VREAP suitable, satisfying, and attractive. Participants notably commented on the positive nature of their experience during training, which mirrors reactions to VR in feasibility studies that focused on VR gaming for physical activity [51]. Participants further noted the usefulness of the VREAP, with many commenting that the training helped increase their understanding of how VR can be used in sport. These findings are supported by research on VR and XR in exercise settings, where a recent scoping review found that VR immersion in both game-like and realistic environments was positively associated with user motivation and enjoyment [8].

Perceived demand of the VREAP was also evident, as several participants believed it could or should be used in applied settings. However, a barrier to its demand in sport could be coach and practitioner perceptions of VR. In one study with European soccer coaches, barriers to coach adoption were concerns about the practicality of VR, a lack of empirical evidence for VR's value in sport, the quality of VR, and the risk of cognitive overload for players using VR [22]. Another study showed that the most important factor affecting coach and practitioner use of VR training in their sport was whether it could improve on-field performance [23]. Mascret et al. also found that coaches' perceived usefulness and perceived enjoyment of VR for athletes were positive predictors of the coaches' intention to use VR [52], which underscores the possibility that coaches who do not perceive VR as useful for their athletes may be more hesitant to adopt it. In short, while athletes in the present study perceive a demand for the VREAP in sport, coach and practitioner perceptions of VR may limit its actual demand and use.

Implementation of the VREAP included minimal factors that affected the difficulty of implementation (e.g., technical difficulties). The research team's knowledge of how to handle such situations allowed for efficient implementation. This supports findings from one early VR feasibility study where staff spent twice as much time setting up and cleaning up the equipment than in patient teaching or patient participation [53]. Other factors from that study that hindered implementation were the availability of staff and the learning curve for staff to operate the equipment. These findings, combined with those of the current study, underscore the importance of well-trained facilitators who can easily troubleshoot issues when implementing a VR intervention. Of note, three participants in the current study had to remove their eyeglasses to continue use, as their glasses would not comfortably fit while wearing the VR headset. Though these participants did not express that this negatively impacted their experience, it is important for future practitioners and researchers to be aware of such characteristics that could negatively impact user experience. Similar concerns were raised in a recent study examining football players' perceptions of VR use where, among the findings, participants expressed concerns that the VR HMD could be uncomfortable, which could increase the difficulty of use, reduce enjoyment, or cause users to become disengaged [54]. These findings support broader research findings among various populations that such experiences can influence VR acceptance and intent to use it in the future [29–31].

There was minimal variation in the speed and efficiency among the five groups. The small variation in times was dependent on how long each rest period lasted for each group. Strong cohesion within a group appeared to lead to longer conversations and thus longer rest periods. For example, Group 4, a group of teammates, was more conversational and used more rest time than Group 1, a group with no prior relationships. The quality of the VREAP was also evident, as several participants commented that it was set up well and planned in a manner that allowed it to run smoothly. There were also no dropouts or need for outside materials, and participants were able to carry out the training tasks themselves, which furthers shows the efficiency and quality of the VREAP with this sample.

Research has suggested that specific training or exposure protocols designed to acclimate users to VR are needed [20,32]. While designing the VREAP in response to those calls, we incorporated research that suggested new VR users should be initially exposed to a low cognitive workload to minimize the risk of cybersickness [33] and incorporate at least 12 minutes of recovery time between uses [42]. While overall scores on the VRSQ increased in the current study, the VREAP was developed to minimize cybersickness, not prevent it from increasing at all. Additionally, the increases were minimal, not statistically significant, and did not negatively impact participants' experiences, which could suggest the VREAP's success in minimizing cybersickness. Even so, it would be useful to compare scores to those of participants who do not complete the VREAP and compare trends in a future study.

Statistical analyses and the reflexive content analysis also suggested increased acceptance of VR after completing the VREAP. The TAM [26] provides a useful interpretation of these results. Increases in perceived usefulness could be a result of the educational component of the VREAP and the sport-specific application used. Perceptions of ease of use may have been influenced by participants' exposure and acclimation to VR through the VREAP, which allowed them to better understand how to operate VR. Participants also indicated high levels of intent to use VR after the VREAP. These findings are in line with several studies using the TAM that showed perceived usefulness to be the strongest predictor of intention to use technology [27] and that it significantly and positively influences intent to use VR [28]. Within sport, Mascret and colleagues [31] also found perceived usefulness, perceived ease of use, and enjoyment to be predictors of intent to use VR. Participants in the current study also indicated high levels of enjoyment after completing the VREAP, which could be another contributing factor to their expressed intent and willingness to use VR again. While our findings regarding cybersickness and attitudes are preliminary, the demonstrated feasibility of the VREAP among baseball and softball players in the current study shows promise for future research.

## Limitations and future directions

There are several limitations of the current study. First, because the focus of this study was on feasibility, the study design and small sample size limit broad conclusions. Our findings only support the feasibility and potential efficacy of the VREAP, specifically with baseball and softball players. Future research is needed to confirm these findings and test the efficacy of the VREAP using experimental designs with larger sample sizes. Additionally, our sample only included athletes with experience playing baseball and softball. While the VREAP was developed using theories that are proven to span many contexts and domains (e.g., the Technology Acceptance Model and cybersickness) and calls for domain-specific applications to enhance perceived usefulness, we recognize that using a specific population could limit the generalizability of the findings. Future research should evaluate the efficacy of the VREAP using various sports, other domains in which VR has demonstrated usefulness such as medicine and the military [3,6], and user-specific applications in Stage Three (i.e., Application). We also did not have established benchmarks for every feasibility measure because of the limited research on this topic and population. However, the feasibility outcomes in the present study can serve as benchmarks in future research.

One of the primary suggestions from participants for future iterations of the VREAP was to include a bat attachment that would allow them to swing a real bat while using the Win Reality app. We chose not to include the bat attachment for this study due to budget constraints and risk of injury to other participants while swinging. It is possible that the use of a bat could affect participants' perceived usefulness and ease of use, so we suggest future iterations incorporate the use of bats when using Win Reality in Stage Three. It is also possible that participants' positive attitudes toward VR were influenced by the novelty effect (i.e., participants were positively influenced by the excitement of the new technology). Because the current study only assessed attitudes immediately after use, it is possible that the attitudes will change once the novelty of using VR wears off. As such, future research would benefit from conducting follow-up assessments or interviews with participants to assess whether their attitudes change over time.

There are also several variables for which the current study did not control or assess. For example, females may be more likely to experience cybersickness than males [55]. While descriptive statistics in the current study revealed an

increase in scores among females, our results were not statistically significant. Even so, future research should continue to investigate potential gender differences. We also did not control for the total number of previous VR uses. While our inclusion criteria did limit previous uses to five over the previous year, it is possible that participants with zero previous uses could have different attitudes and experiences after the training than someone who had used VR previously. Additionally, we did not assess the possible influence of social norms, which have been shown to positively influence attitudes toward VR [21,31]. Because participants were run in groups, and some participants within those groups knew each other, it is possible that their attitudes and experiences, and thus the results, were influenced by the group's norms. Future research should examine the potential impacts of each of these variables.

The practical potential of the VREAP also provides opportunities for additional research. Within sport, it would be beneficial to examine the use of the VREAP with a team or at a team facility, where implementation may be more difficult than in a controlled laboratory setting. A researcher who subscribes to the scientist-practitioner model may conduct a qualitative case study that describes their experience as a practitioner using the VREAP with a sports team. Researchers should also examine the applied use of the VREAP in other performance domains such as medicine and the military.

While the focus of this study was to assess the feasibility and acceptability of the VREAP among baseball and softball players, there are other potential implications for its practical use. For example, it is also important to use sport- and domain-specific apps during Stage Three so that users with various athletic backgrounds and from different domains can experience and understand how VR can be useful. Moreover, it is important that any practitioners who implement the VREAP are familiar with the technology to ensure a smooth experience for participants. Several athletes also suggested that the applied use of the VREAP should include the use of the Win Reality bat attachment to promote a more realistic feel during use. If baseball or softball athletes, coaches, or practitioners use the VREAP, we suggest using the bat attachment during uses of Win Reality.

Our findings also add to the evolving literature on VR use in baseball and softball. A recent article by Wilkins [56] provided a framework for incorporating VR in baseball. In his Baseball Framework for Applying Virtual Reality (Baseball-FAVR), Wilkins discussed six cases in which VR can be used to enhance psychological performance in baseball based on previous research: pitch recognition training, game day scouting report, pitching strategy, post-game analysis, psychological readiness, and maintaining engagement for injured players. We agree with Wilkins' suggestions and believe that the VREAP and Baseball-FAVR can coexist. For example, if baseball players and coaches have low acceptance of VR, the VREAP could be employed to enhance their acceptance, after which the Baseball-FAVR could be used to improve psychological performance. Such applied implications can be used as a starting point for incorporating VR into practice as research continues to evaluate the efficacy of the VREAP for athletes in various sports and users in other domains.

## Conclusions

Results from this study support the feasibility of the VREAP among baseball and softball players. Specifically, feasibility was demonstrated in recruitment and adherence, as well as acceptability, demand, implementation, and practicality. We also observed positive changes in participant attitudes toward VR after participation and minimal adverse reactions, which suggests that the VREAP may succeed in promoting VR acceptance and minimizing the risk of cybersickness. We encourage the application of the VREAP by athletes, coaches, and practitioners in any performance domain. However, future research should confirm these findings and examine the efficacy of the VREAP for various sports and in various contexts, which will strengthen the support of VR as a training tool, particularly in sport.

## Supporting information

**S1 File. Virtual Reality Education and Acceptance Protocol.**
(DOCX)

## Author contributions

**Conceptualization:** Jarad A. Lewellen, Jeanette M. Garcia, D. Jake Follmer, Guy Hornsby, Peter R. Giacobbi, Jr.

**Data curation:** Jarad A. Lewellen, Cami A. Barnes, Aidan Forget.

**Formal analysis:** Jarad A. Lewellen, Cami A. Barnes, Aidan Forget, Hannah L. Silva-Breen.

**Funding acquisition:** Jarad A. Lewellen.

**Investigation:** Jarad A. Lewellen, Cami A. Barnes, Aidan Forget, Hannah L. Silva-Breen, Peter R. Giacobbi, Jr.

**Methodology:** Jarad A. Lewellen, Jeanette M. Garcia, D. Jake Follmer, Guy Hornsby, Peter R. Giacobbi, Jr.

**Project administration:** Jarad A. Lewellen, Peter R. Giacobbi, Jr.

**Resources:** Jarad A. Lewellen, Peter R. Giacobbi, Jr.

**Supervision:** Jarad A. Lewellen.

**Validation:** Jarad A. Lewellen, Cami A. Barnes, Aidan Forget, Jeanette M. Garcia, D. Jake Follmer, Guy Hornsby, Hannah L. Silva-Breen.

**Visualization:** Jarad A. Lewellen.

**Writing – original draft:** Jarad A. Lewellen, Peter R. Giacobbi, Jr.

**Writing – review & editing:** Jarad A. Lewellen, Cami A. Barnes, Aidan Forget, Jeanette M. Garcia, D. Jake Follmer, Guy Hornsby, Hannah L. Silva-Breen, Peter R. Giacobbi, Jr.

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
