## [Decision Letter · Decision Letter 0]

14 Aug 2025

Dear Dr. Giacobbi,

Thank you for submitting your manuscript to PLOS ONE. After careful consideration, we feel that it has merit but does not fully meet PLOS ONE’s publication criteria as it currently stands. Therefore, we invite you to submit a revised version of the manuscript that addresses the points raised during the review process.

We look forward to receiving your revised manuscript.

Kind regards,

Job Fransen

Academic Editor

PLOS ONE

Journal Requirements:

[This study was funded by the Stitzel Graduate Student Support Endowment awarded to the first author (JAL) at West Virginia University. The funders had no role in study design, data collection and analysis, decision to publish, or preparation of the manuscript.].

Additional Editor Comments:

Dear authors

Our reviewers have recommended that significant revisions should be made before your manuscript can be considered further. Please consider their recommendations and address them. I will then make a consideration about whether these comments were addressed sufficiently before sending it back out for a new round of revisions.

Sincerely

Job

Reviewers' comments:

Reviewer's Responses to Questions

**Comments to the Author**

1. Is the manuscript technically sound, and do the data support the conclusions?

Reviewer #1: Partly

Reviewer #2: Yes

2. Has the statistical analysis been performed appropriately and rigorously?

Reviewer #1: No

Reviewer #2: Yes

3. Have the authors made all data underlying the findings in their manuscript fully available?

Reviewer #1: Yes

Reviewer #2: Yes

4. Is the manuscript presented in an intelligible fashion and written in standard English?

Reviewer #1: Yes

Reviewer #2: Yes

Reviewer #1: The authors have submitted an overall well-written paper exploring the feasibility and acceptability of VR training in baseball and softball. While this paper has some positives, I have some concerns that need to be addressed prior to being considered for publication.

General comments:

- The introduction is well-written for the most part, but needs more work building on past research, differentiating between different types of extended reality and where VR fits in, and accuracy of cited studies. In addition to this, the rationale for the study is somewhat there, but needs to be clearer as it feeds into the aim.

- It is currently not clear what the Virtual Reality Training Protocol for Sport (VRTPS) is. Apologies if there is confusion, but have you created this protocol? What were the steps taken to create this protocol? Or was this protocol adapted from previous research - if so, it has not been covered in the introduction and needs to be. This is a significant current limitation of the study. How confident are you that this should be called Virtual Reality Training Protocol for Sport, when it is focusing on baseball/softball specifically (very specific types of sport)? I would caution against this title.

- It is not clear why statistical analyses were not used to understand differences. I understand there is a small sample size, but please elaborate more and whether there are non-parametric (or other) statistics you could possibly use.

Minor comments:

Abstract

L19 - I would suggest adding cost here also.

Introduction

L54-55 - This study was not a training study, please amend.

L56 - Although the article title suggest VR, Page and colleagues used 360-video, not VR. Please amend, and I recommend providing a greater overview of VR/Extended Reality (XR) earlier in the introduction to discriminate between VR, 360-video, MR, AR.

L80-83 - There are several recent reviews that have explored VR in sport (training). I recommend engaging more with this literature to support your statements here.

L85 - What evidence? No evidence is cited.

L86-89 - This is a very relevant study to your current study, and I believe it warrants further discussion than a single sentence.

L94 - Is this 80% context-specific? This is a strong statement, and needs further elaboration. If 80% of all VR users experienced cybersickness, very few people would use it.

L130-134 - The rationale here seems very short. I recommend expanding on this more to help set up the aims of the study.

I recommend including this study, or justify the choice to exclude this: Wilkins, L. (2024). A framework for using virtual reality to enhance psychological performance in baseball. Journal of Sport Psychology in Action, 1-16.

Methods

L186 - Please cite multiple multiple studies as you have said 'similar feasibility studies'/

L223 - Please cite the previous literature here - this is vital.

L251-252 - Why did you choose to use the controllers only and not the bat? I am surprised this was not included when testing the acceptability of high level participants. This is a design issue that would have overcome some of the comments raised in the Results (though I think these are important to raise)

Results

L303-325 - Apologies if incorrect for feasibility studies, but I feel this would sit better in the Methods? If this is standard practice for feasibility practices, happy for it to remain here.

L384 - Why have no statistical analyses been done here? I am not sure you can say 'increased minimally' - this would be no significant difference.

Discussion

L418 - Please clarify that this was through interview responses.

L436 - Revise wording to start this sentence.

L438-44 - I was very surprised these studies were not included in the introduction. Please include these and a deeper discussion to help provide background for your study.

L452-456 - Do you think that running the data collection in groups (and the conversational nature among these groups) may have influenced the results?

L493 - I agree that this limits the generalisability - I therefore caution against a general term such as VRTPS if you are acknowledging that it might not be generalised to other sports. As you say in L539 - "future research should confirm these findings and examine the efficacy of the VRTPS for various sports"

L506 - Why did you not provide gender comparisons? This should be an easy and important comparison to run.

Thank you for the opportunity to review and best of luck.

Reviewer #2: To enhance flow I would prefer the Virtual Reality Training Protocol for Sport (VRTPS) section to be positioned before the questionnaire section. Can you clarify the experience level of the four coders please. Can you provide more detail on the content analysis process.

Results are logical, well-presented and appropriate. Strong dicussion and conclusion.

**Do you want your identity to be public for this peer review?** For information about this choice, including consent withdrawal, please see our Privacy Policy

Reviewer #1: No

Reviewer #2: No

---

## [Author Response · Author response to Decision Letter 1]

28 Sep 2025

Author Response: Thank you for bringing this to our attention. We have ensured that our revised submission meets the PLOS ONE style and file naming requirements.

[This study was funded by the Stitzel Graduate Student Support Endowment awarded to the first author (JAL) at West Virginia University. The funders had no role in study design, data collection and analysis, decision to publish, or preparation of the manuscript.].

Author Response: Thank you for bringing this to our attention. We have amended the funding statement as noted (please see above in cover letter).

Author Response: Thank you for this comment. Our Methods section begins with the following ethics statement: “This study was conducted after approval was obtained from the West Virginia University Institutional Review Board (Protocol #2410050857). All participants provided written informed consent before participating in the study.” To the best of our knowledge, this includes the required information noted in the comment. Please advise if there is still information missing.

Additional Editor Comments:

Dear authors

Our reviewers have recommended that significant revisions should be made before your manuscript can be considered further. Please consider their recommendations and address them. I will then make a consideration about whether these comments were addressed sufficiently before sending it back out for a new round of revisions.

Sincerely

Job

Reviewers' comments:

Reviewer's Responses to Questions

Comments to the Author

1. Is the manuscript technically sound, and do the data support the conclusions?

Reviewer #1: Partly

Reviewer #2: Yes

2. Has the statistical analysis been performed appropriately and rigorously?

Reviewer #1: No

Reviewer #2: Yes

3. Have the authors made all data underlying the findings in their manuscript fully available?

Reviewer #1: Yes

Reviewer #2: Yes

4. Is the manuscript presented in an intelligible fashion and written in standard English?

Reviewer #1: Yes

Reviewer #2: Yes

5. Review Comments to the Author

Reviewer #1: The authors have submitted an overall well-written paper exploring the feasibility and acceptability of VR training in baseball and softball. While this paper has some positives, I have some concerns that need to be addressed prior to being considered for publication.

Author Response: Thank you for your feedback! We have worked to address you commends that have undoubtedly strengthened our paper.

General comments:

- The introduction is well-written for the most part, but needs more work building on past research, differentiating between different types of extended reality and where VR fits in, and accuracy of cited studies. In addition to this, the rationale for the study is somewhat there, but needs to be clearer as it feeds into the aim.

Author Response: Thank you for your comments. We have addressed your comments below and feel that we have incorporated more recent research, provided an extended discussion on the XR spectrum, provided clarification on our cited studies, and expanded the explanation of the rationale.

- It is currently not clear what the Virtual Reality Training Protocol for Sport (VRTPS) is. Apologies if there is confusion, but have you created this protocol? What were the steps taken to create this protocol? Or was this protocol adapted from previous research - if so, it has not been covered in the introduction and needs to be. This is a significant current limitation of the study. How confident are you that this should be called Virtual Reality Training Protocol for Sport, when it is focusing on baseball/softball specifically (very specific types of sport)? I would caution against this title.

Author Response: Thank you for you comments and questions. We did create this protocol, and we have attempted to make that clearer in the manuscript, specifically in the Virtual Reality Education and Acceptance Protocol subsection (beginning on Line 245). We understand your suggestion that we include the development of the protocol in the introduction. However, because it has not been empirically tested prior to this study and the primary aim is to assess its feasibility, we feel it fits best in the methods, though we did provide more context and detail at the end of the introduction (Lines 204-209). After considering your feedback, we also changed the name of the protocol to more accurately reflect its purpose. We chose to keep the title general, as we intend for the VREAP to be adapted to various domains in future research and application, though we specified that it was adapted to use with baseball and softball players in the title and throughout the manuscript. We have also included more research from other domains to provide more justification for the general name of the protocol.

- It is not clear why statistical analyses were not used to understand differences. I understand there is a small sample size, but please elaborate more and whether there are non-parametric (or other) statistics you could possibly use.

Author Response: Thank you for this comment. After considering reviewer feedback, we have conducted nonparametric tests (i.e., Wilcoxon signed-ranked test) on the AVRTS and VRSQ pre- and post-test means and incorporated the outputs into the results and discussion sections as appropriate. Because the subsection of the results that focused on the VRSQ and AVRTS now includes these analyses (i.e., more than just descriptive trends), we have renamed the section “Cybersickness and attitudes toward VR” (Line 486).

Minor comments:

Abstract

L19 - I would suggest adding cost here also.

Author Response: Thank you for your suggestion. We have added cost to the end of this sentence (see Line 30).

Introduction

L54-55 - This study was not a training study, please amend.

Author Response: Thank you for bringing that to our attention. We have amended the sentence to remove the suggestion that VR was used for training in this study (see Line 79).

L56 - Although the article title suggest VR, Page and colleagues used 360-video, not VR. Please amend, and I recommend providing a greater overview of VR/Extended Reality (XR) earlier in the introduction to discriminate between VR, 360-video, MR, AR.

Author Response: Thank you for this suggestion. We have included an overview of XR, VR, AR, and MR early in the discussion (Lines 49-61). We have also noted the difference between animated VR and 360-video using a VR headset and specified that the Page et al. study used 360-video (Line 82).

L80-83 - There are several recent reviews that have explored VR in sport (training). I recommend engaging more with this literature to support your statements here.

Author Response: Thank you for this suggestion. We have included findings from two recent reviews (Cariati et al., 2025; Witte et al., 2025) to support our statements (Lines 106-115).

L85 - What evidence? No evidence is cited.

Author Response: Thank you for the note. We have reworded the sentence for clarity and included a recent study that demonstrates a lack of VR use in collegiate sport (Lines 116-119).

L86-89 - This is a very relevant study to your current study, and I believe it warrants further discussion than a single sentence.

Author Response: Thank you for this suggestion. We have included a paragraph with more detail on this study and its findings (Lines 123-137).

L94 - Is this 80% context-specific? This is a strong statement, and needs further elaboration. If 80% of all VR users experienced cybersickness, very few people would use it.

Author Response: Thank you for this note. We decided to remove this sentence from the manuscript as we felt the previous sentence and reference (Lewellen et al., 2025) was sufficient evidence that cybersickness is a barrier. Additionally, after a more careful review of the cybersickness literature, it is difficult to ascertain an accurate percentage of cybersickness likelihood, as there is great variation between studies, and many of them are intentionally eliciting symptoms, which does not accurately reflect the possibility of cybersickness when symptoms are not induced.

L130-134 - The rationale here seems very short. I recommend expanding on this more to help set up the aims of the study.

Author Response: Thank you for this recommendation. We elaborated on the rationale and purpose of the study in Lines 203-214.

I recommend including this study, or justify the choice to exclude this: Wilkins, L. (2024). A framework for using virtual reality to enhance psychological performance in baseball. Journal of Sport Psychology in Action, 1-16.

Author Response: Thank you for this recommendation. We chose not to include this study in the introduction because it was not experimental and thus did not support our rationale and aims. However, we agree it is an important study, especially as it relates to the applied implications of our study, so we included it in our discussion of future directions (Lines 666-677).

Methods

L186 - Please cite multiple studies as you have said 'similar feasibility studies'/

Author Response: Thank you for this note. After careful review, we felt the systematic review that we originally referenced was not an accurate enough comparison to our study and was not necessary to reference. Instead, we felt that concluding this paragraph with the adherence rate from Ross-Stewart et al.’s (2018) study created better flow. We also changed “engagement” to “adherence” in Line 322 to accurately reflect the language used in Ross-Stewart et al. (2018).

L223 - Please cite the previous literature here - this is vital.

Author Response: Thank you for this suggestion. We referenced the relevant previous literature throughout this section as each stage was discussed. We added a sentence explaining this for clarity (Line 248). We also added additional references to support Stages 1 and 3 (Lines 260; 284).

L251-252 - Why did you choose to use the controllers only and not the bat? I am surprised this was not included when testing the acceptability of high level participants. This is a design issue that would have overcome some of the comments raised in the Results (though I think these are important to raise)

Author Response: Thank you for the question. There were two primary reasons we did use bats. 1) We had budget constraints and could not afford to purchase bats or bat attachments for this study. Given that participants did not have to be actively participating in the sport, we also could not rely on them to bring their own bats. 2) Because participants were run in groups, there were four participants using VR simultaneously in the same room. Given the space, we did not want to risk potential injury to participants with swinging, and possibly accidentally letting go of, the bats. Because we understand this could be an important limitation, we also added this context in the Discussion (Lines 627-628).

Results

L303-325 - Apologies if incorrect for feasibility studies, but I feel this would sit better in the Methods? If this is standard practice for feasibility practices, happy for it to remain here.

Author Response: Thank you for the suggestion. While we understand why it might flow better in the Methods, it is standard practice to include this in the results because successful/unsuccessful recruitment methods are a result themselves in a feasibility study. Please see similarly designed feasibility studies below as examples:

• Garcia JM, Shurack R, Leahy N, Brazendale K, Lee E, Lawrence S. Feasibility of a remote-based nutrition education and culinary skills program for young adults with Autism Spectrum Disorder. J Nutr Educ Behav. 2023;55(3):215-23.

• Sandgren SS, Haycraft E, Arcelus J, Plateau CR. Evaluating a motivational and psycho-educational self-help intervention for athletes with mild eating disorder symptoms: A mixed methods feasibility study. Eur Eat Disord Rev. 2022;30(3):250-66.

L384 - Why have no statistical analyses been done here? I am not sure you can say 'increased minimally' - this would be no significant difference.

Author Response: Thank you for this question. Initially, we did not include parametric tests because the primary focus of the study was to assess feasibility. However, as noted above, after considering reviewer feedback, we have conducted nonparametric tests (i.e., Wilcoxon signed-rank test) on the AVRTS and VRSQ pre- and post-test means and incorporated the outputs into the results and discussion sections as appropriate.

Discussion

L418 - Please clarify that this was through interview responses.

Author Response: Thank you for the suggestion. We included this in the sentence (Line 537). We also included that statistical analyses suggest potential of the VREAP, as results from the surveys and the Wilcoxon signed-rank test indicated high acceptance and low cybersickness.

L436 - Revise wording to start this sentence.

Author Response: Thank you for bringing this to our attention. The sentence has been revised.

L438-44 - I was very surprised these stud

---

## [Decision Letter · Decision Letter 1]

23 Oct 2025

Dear Dr. Giacobbi,

Thank you for submitting your manuscript to PLOS ONE. After careful consideration, we feel that it has merit but does not fully meet PLOS ONE’s publication criteria as it currently stands. Therefore, we invite you to submit a revised version of the manuscript that addresses the points raised during the review process.

We look forward to receiving your revised manuscript.

Kind regards,

Job Fransen

Academic Editor

PLOS ONE

Journal Requirements:

Additional Editor Comments:

Dear authors

Reviewer one has some minor changes they would like to see being addressed before we can proceed.

Reviewers' comments:

Reviewer's Responses to Questions

**Comments to the Author**

Reviewer #1: (No Response)

Reviewer #2: All comments have been addressed

2. Is the manuscript technically sound, and do the data support the conclusions?

Reviewer #1: Yes

Reviewer #2: Yes

3. Has the statistical analysis been performed appropriately and rigorously?

Reviewer #1: Yes

Reviewer #2: Yes

4. Have the authors made all data underlying the findings in their manuscript fully available?

Reviewer #1: Yes

Reviewer #2: Yes

5. Is the manuscript presented in an intelligible fashion and written in standard English?

Reviewer #1: Yes

Reviewer #2: Yes

Reviewer #1: Well done to the authors for their thorough revisions of this manuscript. I appreciate that a long list of revisions can be a challenge, but the authors have done a thorough job. One recommendation I will make to the authors is to have any changes made to the revised manuscript in red font. This would make it easier to review and clearly identify where changed were made.

I appreciate the authors' changes to the title and think that this is a better title for the study.

My main suggestion is for parts of the discussion to be expanded to really tie together the paper's findings. The main parts to focus on are the paragraphs from Line 552 to Line 576. I think you could draw on research in sport using VR (and other XR technologies) to highlight whether these have been more game like, enjoyment, difficult to use. You could also consider bringing in work from other domains such as education, medical training.

Well done again and best of luck. I look forward to hopefully applying this protocol in my own XR research.

Reviewer #2: The study is interetsing. The use of 360 and VR is still intermittant across sports, so the findings will add value to the discussions in the area.

**Do you want your identity to be public for this peer review?** For information about this choice, including consent withdrawal, please see our Privacy Policy

Reviewer #1: **Yes: ** Aden Kittel

Reviewer #2: **Yes: ** Kirsten Spencer

---

## [Author Response · Author response to Decision Letter 2]

31 Oct 2025

Manuscript PONE-D-25-29500

Response to Reviewers

Dear Dr. Fransen and Reviewers,

Thank you for your additional feedback on our manuscript. We have taken all comments into consideration and provided our responses in blue below. Additionally, the changes are highlighted in the Revised Manuscript with Track Changes file. In this file, Per Reviewer #1’s request, we have also marked the additional changes to the manuscript in red font to make it easier to identify where changes have been made.

Please also note that Microsoft Word has a glitch that changes the line numbers when Track Changes is set to “All Markup,” and that our line numbers below reflect the accurate continuous line numbers in the Manuscript file with no track changes, or when the track changes function is set to “Simple Markup” in the track changes file.

Best,

Jarad A. Lewellen and Peter R. Giacobbi, Jr.

Journal Requirements:

Author Response: Thank you for these comments. To the best of our knowledge, none of our cited papers have been retracted. We have also reviewed our reference list to ensure accuracy (e.g., reviewed formatting and included DOIs).

Dear authors

Reviewer one has some minor changes they would like to see being addressed before we can proceed.

Reviewers' comments:

Reviewer's Responses to Questions

Comments to the Author

1. If the authors have adequately addressed your comments raised in a previous round of review and you feel that this manuscript is now acceptable for publication, you may indicate that here to bypass the “Comments to the Author” section, enter your conflict of interest statement in the “Confidential to Editor” section, and submit your "Accept" recommendation.

Reviewer #1: (No Response)

Reviewer #2: All comments have been addressed

2. Is the manuscript technically sound, and do the data support the conclusions?

Reviewer #1: Yes

Reviewer #2: Yes

3. Has the statistical analysis been performed appropriately and rigorously?

Reviewer #1: Yes

Reviewer #2: Yes

4. Have the authors made all data underlying the findings in their manuscript fully available?

Reviewer #1: Yes

Reviewer #2: Yes

5. Is the manuscript presented in an intelligible fashion and written in standard English?

Reviewer #1: Yes

Reviewer #2: Yes

6. Review Comments to the Author

Reviewer #1: Well done to the authors for their thorough revisions of this manuscript. I appreciate that a long list of revisions can be a challenge, but the authors have done a thorough job. One recommendation I will make to the authors is to have any changes made to the revised manuscript in red font. This would make it easier to review and clearly identify where changed were made.

Author Response: Thank you for your kind comments and suggestion. We have marked all changes to this version of revisions in red font (in the Revised Manuscript with Track Changes file) to make the changes easier to identify.

I appreciate the authors' changes to the title and think that this is a better title for the study.

My main suggestion is for parts of the discussion to be expanded to really tie together the paper's findings. The main parts to focus on are the paragraphs from Line 552 to Line 576. I think you could draw on research in sport using VR (and other XR technologies) to highlight whether these have been more game like, enjoyment, difficult to use. You could also consider bringing in work from other domains such as education, medical training.

Author Response: Thank you for these suggestions. We have expanded on the content in these paragraphs to tie our findings together with other research. Specifically, we included references on game-like and realistic experiences among exercise populations, impacts of perceived usefulness and enjoyment on demand among coaches, and potential impact of eyeglass use and HMD discomfort on VR experience and acceptance (Lines 557-589).

Well done again and best of luck. I look forward to hopefully applying this protocol in my own XR research.

Author Response: Thank you for this comment and for taking the taking the time to review our manuscript!

Reviewer #2: The study is interesting. The use of 360 and VR is still intermittent across sports, so the findings will add value to the discussions in the area.

Author Response: Thank you for this feedback and for taking the time to review our manuscript!

---

## [Decision Letter · Decision Letter 2]

10 Nov 2025

Assessing the feasibility of the Virtual Reality Education and Acceptance Protocol among baseball and softball players

PONE-D-25-29500R2

Dear Dr. Giacobbi,

We’re pleased to inform you that your manuscript has been judged scientifically suitable for publication and will be formally accepted for publication once it meets all outstanding technical requirements.

Kind regards,

Job Fransen

Academic Editor

PLOS ONE

Additional Editor Comments (optional):

Reviewers' comments:

Reviewer's Responses to Questions

**Comments to the Author**

Reviewer #1: All comments have been addressed

2. Is the manuscript technically sound, and do the data support the conclusions?

Reviewer #1: Yes

3. Has the statistical analysis been performed appropriately and rigorously?

Reviewer #1: Yes

4. Have the authors made all data underlying the findings in their manuscript fully available?

Reviewer #1: Yes

5. Is the manuscript presented in an intelligible fashion and written in standard English?

Reviewer #1: Yes

Reviewer #1: Well done on this paper. I do not have any further suggestions and recommend publication. Best of luck for future work.

**Do you want your identity to be public for this peer review?** For information about this choice, including consent withdrawal, please see our Privacy Policy

Reviewer #1: No

---

## [Editor Report · Acceptance letter]

PONE-D-25-29500R2

PLOS ONE

Dear Dr. Giacobbi, Jr.,

I'm pleased to inform you that your manuscript has been deemed suitable for publication in PLOS ONE. Congratulations! Your manuscript is now being handed over to our production team.

Kind regards,

on behalf of

Dr. Job Fransen

Academic Editor

PLOS ONE